# On Robustness of Kernel Clustering

**Bowei Yan**
Department of Statistics and Data Sciences
University of Texas at Austin

**Purnamrita Sarkar**
Department of Statistics and Data Sciences
University of Texas at Austin

## Abstract

Clustering is an important unsupervised learning problem in machine learning
and statistics. Among many existing algorithms, kernel k-means has drawn much
research attention due to its ability to find non-linear cluster boundaries and its
inherent simplicity. There are two main approaches for kernel k-means: SVD
of the kernel matrix and convex relaxations. Despite the attention kernel cluster-
ing has received both from theoretical and applied quarters, not much is known
about robustness of the methods. In this paper we first introduce a semidefinite
programming relaxation for the kernel clustering problem, then prove that under a
suitable model specification, both K-SVD and SDP approaches are consistent in
the limit, albeit SDP is strongly consistent, i.e. achieves exact recovery, whereas
K-SVD is weakly consistent, i.e. the fraction of misclassified nodes vanish. Also
the error bounds suggest that SDP is more resilient towards outliers, which we also
demonstrate with experiments.

## 1 Introduction

Clustering is an important problem which is prevalent in a variety of real world problems. One of the
first and widely applied clustering algorithms is k-means, which was named by James MacQueen [14],
but was proposed by Hugo Steinhaus [21] even before. Despite being half a century old, k-means has
been widely used and analyzed under various settings.

One major drawback of k-means is its incapability to separate clusters that are non-linearly separated.
This can be alleviated by mapping the data to a high dimensional feature space and do clustering on
top of the feature space [19, 9, 12], which is generally called kernel-based methods. For instance,
the widely-used spectral clustering [20, 16] is an algorithm to calculate top eigenvectors of a kernel
matrix of affinities, followed by a k-means on the top $r$ eigenvectors. The consistency of spectral
clustering is analyzed by [22]. [9] shows that spectral clustering is essentially equivalent to a weighted
version of kernel k-means.

The performance guarantee for clustering is often studied under distributional assumptions; usually
a mixture model with well-separated centers suffices to show consistency. [5] uses a Gaussian
mixture model, and proposes a variant of EM algorithm that provably recovers the center of each
Gaussian when the minimum distance between clusters is greater than some multiple of the square
root of dimension. [2] works with a projection based algorithm and shows the separation needs to
be greater than the operator norm and the Frobenius norm of difference between data matrix and its
corresponding center matrix, up to a constant.

Another popular technique is based on semidefinite relaxations. For example [18] proposes a SDP
relaxation for k-means typed clustering. In a very recent work, [15] shows the effectiveness of SDP
relaxation with k-means clustering for subgaussian mixtures, provided the minimum distance between
centers is greater than the variance of the sub-gaussian times the square of the number of clusters $r$.

On a related note, SDP relaxations have been shown to be consistent for community detection in
networks [1, 3]. In particular, [3] consider "inlier" (these are generated from the underlying clustering

model, to be specific, a blockmodel) and "outlier" nodes. The authors show that SDP is weakly consistent in terms of clustering the inlier nodes as long as the number of outliers $m$ is a vanishing fraction of the number of nodes.

In contrast, among the numerous work on clustering, not much focus has been on robustness of different kernel k-means algorithms in presence of arbitrary outliers. [24] illustrates the robustness of Gaussian kernel based clustering, where no explicit upper bound is given. [8] detects the influential points in kernel PCA by looking at an influence function. In data mining community, many find clustering can be used to detect outliers, with often heuristic but effective procedures [17, 10]. On the other hand, kernel based methods have been shown to be robust for many machine learning tasks. For supervised learning, [23] shows the robustness of SVM by introducing an outlier indicator and relaxing the problem to a SDP. [6, 7, 4] develop the robustness for kernel regression. For unsupervised learning, [13] proposes a robust kernel density estimation.

In this paper we ask the question: how robust are SVD type algorithms and SDP relaxations when outliers are present. In the process we also present results which compare these two methods. To be specific, we show that without outliers, SVD is weakly consistent, i.e. the *fraction* of misclassified nodes vanishes with high probability, whereas SDP is strongly consistent, i.e. the *number* of misclassified nodes vanishes with high probability. We also prove that both methods are robust to arbitrary outliers as long as the number of outliers is growing at a slower rate than the number of nodes. Surprisingly our results also indicate that SDP relaxations are more resilient to outliers than K-SVD methods. The paper is organized as follows. In Section 2 we set up the problem and the data generating model. We present the main results in Section 3. Proof sketch and more technical details are introduced in Section 4. Numerical experiments in Section 5 illustrate and support our theoretical analysis. More additional analysis are included in the extended version of this paper [1].

## 2    Problem Setup

We denote by $Y = [Y_1, \cdots, Y_n]^T$ the $n \times p$ data matrix. Among the $n$ observations, $m$ outliers are distributed arbitrarily, and $n - m$ inliers form $r$ equal-sized clusters, denoted by $C_1, \cdots, C_r$. Let us denote the index set of inliers by $\mathcal{I}$ and index set of outliers by $\mathcal{O}$, $\mathcal{I} \cup \mathcal{O} = [n]$. Also denote by $\mathcal{R} = \{(i, j) : i \in \mathcal{O} \text{ or } j \in \mathcal{O}\}$.

The problem is to recover the true and unknown data partition given by a membership matrix $Z = \{0, 1\}^{n \times r}$, where $Z_{ik} = 1$ if $i$ belongs to the $k$-th cluster and 0 otherwise. For convenience we assume the outliers are also arbitrarily equally assigned to $r$ clusters, so that each extended cluster, denoted by $\tilde{C}_i, i \in [r]$ has exactly $n/r$ points. A ground truth clustering matrix $X_0 \in \mathbb{R}^{n \times n}$ can be achieved by $X_0 = ZZ^T$. It can be seen that $X_0(i, j) = \begin{cases} 1 & \text{if } i, j \in \mathcal{I}, \text{ and belong to the same cluster;} \\ 0 & \text{otherwise.} \end{cases}$

For the inliers, we assume the following mixture distribution model.

$$\text{Conditioned on } Z_{ia} = 1, \quad Y_i = \mu_a + \frac{W_i}{\sqrt{p}}, \ \mathbb{E}[W_i] = 0, \ Cov[W_i] = \sigma_a^2 I_p,$$

$$W_i \text{ are independent sub-gaussian random vectors.}$$

We treat $Y$ as a low dimensional signal hidden in high dimensional noise. More concretely $\mu_a$ is sparse and $\|\mu_a\|_0$ does not depend on $n$ or $p$; as $n \to \infty$, $p \to \infty$. $W_i$'s for $i \in [n]$ are independent. For simplicity, we assume the noise is isotropic and the covariance only depends on the cluster. The sub-gaussian assumption is non-parametric and includes most of the commonly used distribution such as Gaussian and bounded distributions. We include some background materials on sub-gaussian random variables in Appendix A. This general setting for inliers is common and also motivated by many practical problems where the data lies on a low dimensional manifold, but is obscured by high-dimensional noise [11].

We use the kernel matrix based on Euclidean distances between covariates. Our analysis can be extended to inner product kernels as well. From now onwards, we will assume that the function generating the kernel is bounded and Lipschitz.

**Assumption 1.** *For $n$ observations $Y_1, \cdots, Y_n$, the kernel matrix (sometimes also called Gram matrix) $K$ is induced by $K(i,j) = f(\|Y_i - Y_j\|_2^2)$, where $f$ satisfies $|f(x)| \leq 1, \forall x$ and $\exists C_0 > 0, s.t. \sup_{x,y} |f(x) - f(y)| \leq C_0 |x - y|$.*

A widely used example that satisfies the above condition is the Gaussian kernel. For simplicity, we will without loss of generality assume $K(x,y) = f(\|x - y\|^2) = \exp(-\eta\|x - y\|^2)$.

For the asymptotic analysis, we use the following standard notations for approximated rate of convergence. $T(n)$ is $O(f(n))$ iff for some constant $c$ and $n_0$, $T(n) \leq cf(n)$ for all $n \geq n_0$; $T(n)$ is $\Omega(f(n))$ if for some constant $c$ and $n_0$, $T(n) \geq cf(n)$ for all $n \geq n_0$; $T(n)$ is $\Theta(f(n))$ if $T(n)$ is $O(f(n))$ and $\Omega(f(n))$; $T(n)$ is $o(f(n))$ if $T(n)$ is $O(f(n))$ but not $\Omega(f(n))$. $T(n)$ is $o_P(f(n))$ ( or $O_P(f(n))$) if it is $o(f(n))$ (or $O(f(n))$) with high probability.

Several matrix norms are considered in this manuscript. Assume $M \in \mathbb{R}^{n \times n}$, the $\ell_1$ and $\ell_\infty$ norm are defined the same as the vector $\ell_1$ and $\ell_\infty$ norm. We define: $\|M\|_1 := \sum_{ij} |M_{ij}|$ and $\|M\|_\infty := \max_{i,j} |M_{ij}|$. For two matrices $M, Q \in \mathbb{R}^{m \times n}$, their inner product is $\langle M, Q \rangle = \text{trace}(M^T Q)$. The operator norm $\|M\|$ is simply the largest singular value of $M$, which equals the largest eigenvalue for a symmetric matrix. Throughout the manuscript, we use $\mathbf{1}_n$ to represent the all one $n \times 1$ vector and $E_n, E_{n,k}$ to represent the all one matrix with size $n \times n$ and $n \times k$. The subscript will be dropped when it is clear from context.

## 2.1 Two kernel clustering algorithms

Kernel clustering algorithms can be broadly divided into two categories; one is based on semidefinite relaxation of the k-means objective function and the other is eigen-decomposition based, like kernel PCA, spectral clustering, etc. In this section we describe these two settings.

**SDP relaxation for kernel clustering** It is well known [9] that kernel k-means could be achieved by maximizing $\text{trace}(Z^T K Z)$ where $Z$ is the $n \times r$ matrix of cluster memberships. However due to the non-convexity of the constraints, the problem is NP-hard. Thus lots of convex relaxations are proposed in literature. In this paper, we propose the following semidefinite programming relaxation. The same relaxation has been used in stochastic block models [1].

$$\max_X \ \text{trace}(KX) \qquad\qquad \text{(SDP-1)}$$
$$\text{s.t.,} \ \ X \succeq 0, X \geq 0, \ X\mathbf{1} = \frac{n}{r}\mathbf{1}, \ \text{diag}(X) = \mathbf{1}$$

The clustering procedure is listed in Algorithm 1.

---
**Algorithm 1** SDP relaxation for kernel clustering
---
**Require:** Observations $Y_1, \cdots, Y_n$, kernel function $f$.
  1: Compute kernel matrix $K$ where $K(i,j) = f(\|Y_j - Y_j\|_2^2)$;
  2: Solve SDP-1 and let $\hat{X}$ be the optimal solution;
  3: Do k-means on the $r$ leading eigenvectors $U$ of $\hat{X}$.
---

**Kernel singular value decomposition** Kernel singular value decomposition (K-SVD) is a spectral based clustering approach. One first do SVD on the kernel matrix, then do k-means on first $r$ eigenvectors. Different variants include K-PCA which uses singular vectors of centered kernel matrix and spectral clustering which uses singular vectors of normalized kernel matrix. The detailed algorithm is shown in Algorithm 2.

# 3 Main results

In this section we summarize our main results. In this paper we analyze SDP relaxation of kernel k-means and K-SVD type methods. Our main contribution is two-fold. First, we show that SDP relaxation produces strongly consistent results, i.e. the number of misclustered nodes goes to zero with high probability when there are no outliers, which means $r$ without rounding. On the other

---

**Algorithm 2** K-SVD (K-PCA, spectral clustering)

---

**Require:** Observations $Y_1, \cdots, Y_n$, kernel function $f$.
 1: Compute kernel matrix $K$ where $K(i,j) = f(\|Y_j - Y_j\|_2^2)$;
 2: **if** K-PCA **then**
 3:   $K = K - K11^T/n - 11^T K/n + 11^T K11^T/n^2$;
 4: **else if** spectral clustering **then**
 5:   $K = D^{-1/2}KD^{-1/2}$ where $D = \mathrm{diag}(K1_n)$;
 6: **end if**
 7: Do k-means on the $r$ leading singular vectors $V$ of $K$.

---

hand, K-SVD is weakly consistent, i.e. fraction of misclassified nodes goes to zero when there are no outliers. In presence of outliers, we see an interesting dichotomy in the behaviors of these two methods. Both can be proven to be weakly consistent in terms of misclassification error. However, SDP is more resilient to the effect of outliers than K-SVD, if the number of clusters grows or if the separation between the cluster means decays.

Our analysis is organized as follows. First we present a result on the concentration of kernel matrices around their population counterpart. The population kernel matrix for inliers is blockwise constant with $r$ blocks (except the diagonal, which is one). Next we prove that as $n$ increases, the optima $\hat{X}$ of SDP-1 converges strongly to $X_0$, when there are no outliers and weakly if the number of outliers grows slowly with $n$. Then we show the mis-clustering error of the clustering returned by Algorithm 1 goes to zero with probability tending to one as $n \to \infty$ when there are no outliers. Finally, when the number of outliers is growing slowly with $n$, the fraction of mis-clustered nodes from algorithms 1 and 2 converges to zero.

We will start with the concentration of kernel matrices to their population counterpart. We show that under our data model (1) the empirical kernel matrix with the Gaussian kernel restricted on inliers concentrates around a "population" matrix $\tilde{K}$, and the $\ell_\infty$ norm of $K_f^{\mathcal{I}\times\mathcal{I}} - \tilde{K}_f^{\mathcal{I}\times\mathcal{I}}$ goes to zero at the rate of $O(\sqrt{\frac{\log p}{p}})$.

**Theorem 1.** *Let $d_{k\ell} = \|\mu_k - \mu_\ell\|$. For $i \in \tilde{C}_k, j \in \tilde{C}_\ell$, define*

$$\tilde{K}_f(i,j) = \begin{cases} f(d_{k\ell}^2 + \sigma_k^2 + \sigma_\ell^2) & \text{if } i \neq j, \\ f(0) & \text{if } i = j. \end{cases} \tag{1}$$

*Then there exists constant $\rho > 0$, such that $P(\|K_f^{\mathcal{I}\times\mathcal{I}} - \tilde{K}_f^{\mathcal{I}\times\mathcal{I}}\|_\infty \geq c\sqrt{\frac{\log p}{p}}) \leq n^2 p^{-\rho c^2}$.*

**Remark 1.** *Setting $c = \sqrt{\frac{3\log n}{p\log p}}$, there exists constant $\rho > 0$, such that*

$$P\left(\|K^{\mathcal{I}\times\mathcal{I}} - \tilde{K}^{\mathcal{I}\times\mathcal{I}}\|_\infty \geq \sqrt{\frac{3\log n}{\rho p}}\right) \leq \frac{1}{n}.$$

*The error probability goes to zero for a suitably chosen constant as long as $p$ is growing faster than $\log n$.*

While our analysis is inspired by [11], there are two main differences. First we have a mixture model where the population kernel is blockwise constant. Second, we obtain $\sqrt{\frac{\log p}{p}}$ *rates* of convergence by carefully bounding the tail probabilities. In order to attain this we further assume that the noise is sub-gaussian and isotropic. From now on we will drop the subscript $f$ and refer to the kernel matrix as $K$.

By definition, $\tilde{K}$ is blockwise constant with $r$ unique rows (except the diagonal elements which are ones). An important property of $\tilde{K}$ is that $\lambda_r - \lambda_{r+1}$ (where $\lambda_i$ is the $i^{th}$ largest eigenvalue of $\tilde{K}$) will be $\Omega(n\lambda_{\min}(B)/r)$. $B$ is the $r \times r$ Gaussian kernel matrix generated by the centers.

**Lemma 1.** *If the scale parameter in Gaussian kernel is non-zero, and none of the clusters shares a same center, let $B$ be the $r \times r$ matrix where $B_{k\ell} = f(\|\mu_k - \mu_\ell\|)$, then*

$$\lambda_r(\tilde{K}) - \lambda_{r+1}(\tilde{K}) \geq \frac{n}{r}\lambda_{\min}(B) \cdot \min_k \left(f(\sigma_k^2)\right)^2 - 2\max_k(1 - f(2\sigma_k^2)) = \Omega(n\lambda_{\min}(B)/r)$$

Now we present our result on the consistency of SDP-1. To this end, we will upper bound $\|\hat{X} - X_0\|_1$, where $\hat{X}$ is the optima returned by SDP-1 and $X_0$ is the true clustering matrix. We first present a lemma, which is crucial to the proof of the theorem. Before presenting this, we define

$$\gamma_{k\ell} := f(2\sigma_k^2) - f(d_{k\ell}^2 + \sigma_k^2 + \sigma_\ell^2); \qquad \gamma_{\min} := \min_{\ell \neq k} \gamma_{k\ell} \qquad (2)$$

The first quantity $\gamma_{k\ell}$ measures separation between the two clusters $k$ and $\ell$. The second quantity measures the smallest separation possible. We will assume that $\gamma_{min}$ is positive. This is very similar to the analysis in asymptotic network analysis where strong assortativity is often assumed. Our results show that the consistency of clustering deteriorates as $\gamma_{\min}$ decreases.

**Lemma 2.** *Let $\hat{X}$ be the solution to (SDP-1), then*

$$\|X_0 - \hat{X}\|_1 \leq \frac{2\langle K - \tilde{K}, \hat{X} - X_0 \rangle}{\gamma_{\min}} \qquad (3)$$

Combining the above with the concentration of $K$ from Theorem 1 we have the following result:

**Theorem 2.** *When $d_{k\ell}^2 > |\sigma_k^2 - \sigma_\ell^2|, \forall k \neq \ell$, and $\gamma_{\min} = \Omega\left(\sqrt{\frac{\log p}{p}}\right)$ then for some absolute constant $c > 0$, $\|X_0 - \hat{X}\|_1 \leq \max\left\{o_P(1), o_P\left(\frac{mn}{r\gamma_{\min}}\right)\right\}$.*

**Remark 2.** *When there's no outlier in the data, i.e., $m = 0$, $\hat{X} = X_0$ with high probability and SDP-1 is strongly consistent without rounding. When $m > 0$, the right hand side of the inequality is dominated by $mn/r$. Note that $\|X_0\|_1 = \frac{n^2}{r}$, therefore after suitable normalization, the error rate goes to zero with rate $O(m/(n\gamma_{\min}))$ when $n \to \infty$.*

Now we will present the mis-clustering error rate of Algorithm 1 and 2. Although $\hat{X}$ is strongly consistent in the absence of outliers, in practice one often wants to get the labeling in addition to the clustering matrix. Therefore it is usually needed to carry out the last eigen-decomposition step in Algorithm 1. Since $X_0$ is the clustering matrix, its principal eigenvectors are blockwise constant. In order to show small mis-clustering error one needs to show that the eigenvectors of $\hat{X}$ are converging (modulo a rotation) to those of $X_0$. This is achieved by a careful application of Davis-Kahan theorem, a detailed discussion of which we defer to the analysis in Section 4.

The Davis-Kahan theorem lets one bound the deviation of the $r$ principal eigenvectors $\hat{U}$ of a Hermitian matrix $\hat{M}$, from the $r$ principal eigenvectors $U$ of $M$ as : $\|\hat{U} - UO\|_F \leq 2^{3/2}\|M - \hat{M}\|_F/(\lambda_r - \lambda_{r+1})$ [25], where $\lambda_r$ is the $r^{th}$ largest eigenvalue of $M$ and $O$ is the optimal rotation matrix. For a complete statement of the theorem see Appendix F.

Applying the result to $X_0$ and $\tilde{K}$ provides us with two different upper bounds on the distance between leading eigenvectors. We will see in Theorem 3 that the eigengap derived by two algorithms differ, which results in different upper bounds for number of misclustered nodes. Since the Davis-Kahan bounds are tight up-to a constant [25], despite being upper bounds, this indicates that algorithm 1 is less sensitive to the separation between cluster means than Algorithm 2.

Once the eigenvector deviation is established, we present explicit bounds on mis-clustering error for both methods in the following theorem. K-means assigns each row of $\hat{U}$ (input eigenvectors of $K$ or $\hat{X}$) to one of $r$ clusters. Define $c_1 \cdots, c_n \in \mathbb{R}^r$ such that $c_i$ is the centroid corresponding to the $i^{th}$ row of $\hat{U}$. Similarly, for the population eigenvectors $U$ (top $r$ eigenvectors of $\tilde{K}$ or $X_0$), we define population centroids as $(Z\nu)_i$, for some $\nu \in \mathbb{R}^{r \times r}$. Recall that we construct $Z$ such that the outliers are equally and arbitrarily divided amongst the $r$ clusters. We show that when the empirical centroids are close to the population centroids with a rotation, then the node will be correctly clustered. We give a general definition of a superset of the misclustered nodes applicable both to K-SVD and SDP:

$$\mathcal{M} = \{i : \|c_i - Z_i\nu O\| \geq 1/\sqrt{2n/r}\} \qquad (4)$$

**Theorem 3.** *Let $\mathcal{M}_{sdp}$ and $\mathcal{M}_{ksvd}$ be defined as Eq. 4, where $c_i$'s are generated from Algorithm 1 and 2 respectively. Let $\lambda_r$ be the $r^{th}$ largest eigenvalue value of $\tilde{K}$. We have:*

$$|\mathcal{M}_{sdp}| \leq \max\left\{o_P(1), O_P\left(\frac{m}{\gamma_{\min}}\right)\right\}$$

$$|\mathcal{M}_{ksvd}| \leq O_P \max\left\{\frac{mn^2}{r(\lambda_r - \lambda_{r+1})^2}, \frac{n^3 \log p}{rp(\lambda_r - \lambda_{r+1})^2}\right\}$$

**Remark 3.** *Getting a bound for $\lambda_r$ in terms of $\gamma_{\min}$ for general blockwise constant matrices is difficult. But as shown in Lemma 1, the eigengap is $\Omega(n/r\lambda_{min}(B))$. Plugging this back in we have,*

$$|\mathcal{M}_{ksvd}| \leq \max \left\{ O_P \left( \frac{mr}{\lambda_{\min}(B)^2} \right), O_P \left( \frac{nr \log p/p}{\lambda_{\min}(B)^2} \right) \right\}$$

.

In some simple cases one can get explicit bounds for $\lambda_r$, and we have the following.

**Corollary 1.** *Consider the special case when all clusters share the same variance $\sigma^2$ and $d_{k\ell}$ are identical for all pairs of clusters. The number of misclustered nodes of K-SVD is upper bounded by:*

$$|\mathcal{M}_{ksvd}| \leq \max \left( O_P \left( \frac{mr}{\gamma_{\min}^2} \right), O_P \left( \frac{nr \log p/p}{\gamma_{\min}^2} \right) \right) \tag{5}$$

Corollary 1 is proved in Appendix H.

**Remark 4.** *The situation may happen if cluster center for $a$ is of the form $ce_a$ where $e_a$ is a binary vector with $e_a(i) = \mathbf{1}_{a=i}$. In this case, the algorithm is weakly consistent (fraction of misclassified nodes vanish) when $\gamma_{\min} = \Omega \left( \max\{ \sqrt{\frac{r \log p}{p}}, \sqrt{mr/n} \} \right)$. Compared to $|\mathcal{M}_{sdp}|$, $|\mathcal{M}_{ksvd}|$ an additional factor of $\frac{r}{\gamma_{\min}}$. With same $m, n$, the algorithm has worse upper bound of errors and is more sensitive to $\gamma_{\min}$, which depends both on the data distribution and the scale parameter of the kernel. The proposed SDP can be seen as a denoising procedure which enlarges the separation. It succeeds as long as the denoising is faithful, which requires much weaker assumptions.*

# 4 Proof of the main results

In this section, we show the proof sketch of the main theorems. The full proofs are deferred to supplementary materials.

## 4.1 Proof of Theorem 1

In Theorem 1, we show that if the data distribution is sub-gaussian, the $\ell_\infty$ norm of $K - \tilde{K}$ restricted on the inlier nodes concentrates with rate $O\left( \sqrt{\frac{\log p}{p}} \right)$.

*Proof sketch.* With the Lipschitz condition, it suffices to show $\|Y_i - Y_j\|_2^2$ concentrates to $d_{k\ell}^2 + \sigma_k^2 + \sigma_\ell^2$. To do this, we decompose $\|Y_i - Y_j\|_2^2 = \|\mu_k - \mu_\ell\|_2^2 + 2 \frac{(W_i - W_j)^T}{\sqrt{p}} (\mu_k - \mu_\ell) + \frac{\|W_i - W_j\|_2^2}{p}$. Now it suffices to show the third term concentrates to $\sigma_k^2 + \sigma_\ell^2$ and the second term concentrates around 0. Note the fact that $W_i - W_j$ is sub-gaussian, its square is sub-exponential. With sub-gaussian tail bound and a Bernstein type inequality for sub-exponential random variables, we prove the result. $\square$

With the elementwise bound, the Frobenius norm of the matrix difference is just one more factor of $n$.

**Corollary 2.** *With probability at least $1 - n^2 p^{-\rho c^2}$, $\|K^{\mathcal{I} \times \mathcal{I}} - \tilde{K}^{\mathcal{I} \times \mathcal{I}}\|_F \leq cn\sqrt{\log p/p}$.*

## 4.2 Proof of Theorem 2

Lemma 2 is proved in Appendix D, where we make use of the optimality condition and the constraints in SDP-1. Equipped with Lemma 2 we're ready to prove Theorem 2.

*Proof sketch.* In the outlier-free ideal scenario, Lemma 2 along with the dualtiy of $\ell_1$ and $\ell_\infty$ norms we get $\|\hat{X} - X_0\|_1 \leq \frac{2\|K - \tilde{K}\|_\infty \|\hat{X} - X_0\|_1}{\gamma_{\min}}$. Then by Theorem 1, we get the strong consistency result. When outliers are present, we have to derive a slightly different upper bound. The main idea is to divide the matrices into two parts, one corresponding to the rows and columns of inliers, and the other corresponding to those of the outliers. Now by the concentration result (Theorem 1) on $K$ along with the fact that both the kernel function and $X_0, \hat{X}$ are bounded by 1; and the rows of $\hat{X}$ sums to $n/r$ because of the constraint in SDP-1, we obtain the proof. The full proof is deferred to Appendix E. $\square$

### 4.3 Proof of Theorem 3

Although Theorem 2 provides insights on how close the recovered matrix $\hat{X}$ is to the ground truth, it remains unclear how the final clustering result behaves. In this section, we bound the number of misclassified points by bounding the distance in eigenvectors of $\hat{X}$ and $X_0$. We start by presenting a lemma that provides a bound for k-means step.

K-means is a non-convex procedure and is usually hard to analyze directly. However, when the centroids are well-separated, it is possible to come up with sufficient conditions for a node to be correctly clustered. When the set of misclustered nodes is defined as Eq. 4, the cardinality of $\mathcal{M}$ is directly upper bounded by the distance between eigenvectors. To be explicit, we have the following lemma. Here $\hat{U}$ denotes top $r$ eigenvectors of $K$ for K-SVD and $\hat{X}$ for SDP. $U$ denotes the top $r$ eigenvectors of $\tilde{K}$ for K-SVD and $X_0$ for SDP. $O$ denotes the corresponding rotation that aligns the empirical eigenvectors to their population counterpart.

**Lemma 3.** $\mathcal{M}$ *is defined as Eq.* (4)*, then* $|\mathcal{M}| \leq \frac{8n}{r}\|\hat{U} - UO\|_F^2$.

Lemma 3 is proved in Appendix G.

**Analysis of** $|\mathcal{M}_{\mathbf{sdp}}|$**:** In order to get the deviation in eigenvectors, note the $r^{th}$ eigenvalue of $X_0$ is $n/r$, and $r+1^{th}$ is 0, let $U \in \mathbb{R}^{n \times r}$ be top $r$ eigenvectors of $X$ and $\hat{U}$ be eigenvectors of $X_0$. By applying Davis-Kahan Theorem, we have

$$\exists O, \|\hat{U} - UO\|_F \leq \frac{2^{3/2}\|\hat{X} - X_0\|_F}{n/r} \leq \frac{\sqrt{8\|\hat{X} - X_0\|_1}}{n/r} = O_P\left(\sqrt{\frac{mr}{n\gamma_{\min}}}\right) \qquad (6)$$

Applying Lemma 3,

$$|\mathcal{M}_{sdp}| \leq \frac{8n}{r}\left(\frac{2^{3/2}\|\hat{X} - X_0\|_F}{n/r}\right)^2 \leq \frac{cn}{r}\left(\sqrt{\frac{mr}{n\gamma_{\min}}}\right)^2 \leq O_P\left(\frac{m}{\gamma_{\min}}\right)$$

**Analysis of** $|\mathcal{M}_{\mathbf{ksvd}}|$**:** In the outlier-present kernel scenario, by Corollary 2,

$$\|K - \tilde{K}\|_F \leq \|K^{\mathcal{I} \times \mathcal{I}} - \tilde{K}^{\mathcal{I} \times \mathcal{I}}\|_F + \|K^{\mathcal{R}} - \tilde{K}^{\mathcal{R}}\|_F = O_P(n\sqrt{\log p/p}) + O_P(\sqrt{mn})$$

Again by Davis-Kahan theorem, and the eigengap between $\lambda_r$ and $\lambda_{r+1}$ of $\tilde{K}$ from Lemma 1, let $U$ be the matrix with rows as the top $r$ eigenvectors of $\tilde{K}$. Let $\hat{U}$ be its empirical counterpart.

$$\exists O, \|\hat{U} - UO\|_F \leq \frac{2^{3/2}\|K - \tilde{K}\|_F}{\lambda_r - \lambda_{r+1}} \leq O_P\left(\frac{\max\{\sqrt{mn}, n\sqrt{\log p/p}\}}{\lambda_r - \lambda_{r+1}}\right) \qquad (7)$$

Now we apply Lemma 3 and get the upper bound for number of misclustered nodes for K-SVD.

$$|\mathcal{M}_{ksvd}| \leq \frac{8n}{r}\left(\frac{2^{3/2}C\max\{\sqrt{mn}, n\sqrt{\log p/p}\}}{\lambda_r(\tilde{K}) - \lambda_{r+1}(\tilde{K})}\right)^2$$

$$\leq \frac{Cn}{r}\max\left\{\left(\frac{\sqrt{mn}}{\lambda_r - \lambda_{r+1}}\right)^2, \frac{n^2\log p}{p(\lambda_r - \lambda_{r+1})}\right\}$$

$$\leq O_P\max\left\{\frac{mn^2}{r(\lambda_r - \lambda_{r+1})^2}, \frac{n^3\log p}{rp(\lambda_r - \lambda_{r+1})^2}\right\}$$

## 5 Experiments

In this section, we collect some numerical results. For implementation of the proposed SDP, we use Alternating Direction Method of Multipliers that is used in [1]. In each synthetic experiment, we generate $n - m$ inliers from $r$ equal-sized clusters. The centers of the clusters are sparse and hidden in a $p$-dim noise. For each generated data set, we add in $m$ observations of outliers. To capture the

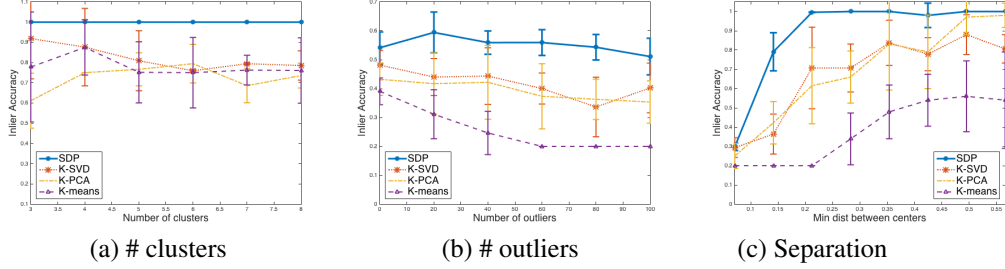

|                     |                    |                     |
|:-------------------:|:------------------:|:-------------------:|
| (a) # clusters      | (b) # outliers     | (c) Separation      |

Figure 1: Performance vs parameters: (a) Inlier accuracy vs number of cluster ($n = p = 1500, m = 10, d^2 = 0.125, \sigma = 1$); (b) Inlier accuracy vs number of outliers ($n = 1000, r = 5, d^2 = 0.02, \sigma = 1, p = 500$); (c) Inlier accuracy vs separation ($n = 1000, r = 5, m = 50, \sigma = 1, p = 1000$).

arbitrary nature of the outliers, we generate half the outliers by a random Gaussian with large variance (100 times of the signal variance), and the other half by a uniform distribution that scatters across all clusters. We compare Algorithm 1 with 1) k-means by Lloyd's algorithms; 2) kernel SVD and 3) kernel PCA by [19].

The evaluating metric is accuracy of inliers, i.e., number of correctly clustered nodes divided by the total number of inliers. To avoid the identification problem, we compare all permutations of the predicted labels to ground truth labels and record the best accuracy. Each set of parameter is run 10 replicates and the mean accuracy and standard deviation (shown as error bars) are reported. For all k-means used in the experiments we do 10 restarts and choose the one with smallest k-means loss.

For each experiment, we change only one parameter and fix all the others. Figure 1 shows how the performance of different clustering algorithms change when (a) number of clusters, (b) number of outliers, (c) minimum distance between clusters, increase. The value of all parameters used are specified in the caption of the figure. Panel (a) shows the inlier accuracy for various methods as we increase number of clusters. It can be seen that with $r$ growing, the performance of all methods deteriorate except for the SDP. We also examine the $\ell_1$ norm of $X_0 - \hat{X}$, which remains stable as the number of clusters increases. Panel (b) describes the trend with respect to number of outliers. The accuracy of SDP on inliers is almost unaffected by the number of outliers while other methods suffer with large $m$. Panel (c) compares the performance as the minimum distance between cluster centers changes. Both SDP and K-SVD are consistent as the distance increases. Compared with K-SVD, SDP achieves consistency faster and variates less across random runs, which matches the analysis given in Section 3.

# 6 Conclusion

In this paper, we investigate the consistency and robustness of two kernel-based clustering algorithms. We propose a semidefinite programming relaxation which is shown to be strongly consistent without outliers and weakly consistent in presence of arbitrary outliers. We also show that K-SVD is also weakly consistent in that the misclustering rate is going to zero as the observation grows and the outliers are of a small fraction of inliers. By comparing two methods, we conclude that although both are robust to outliers, the proposed SDP is less sensitive to the minimum separation between clusters. The experimental result also supports the theoretical analysis.

## Footnotes

[1] https://arxiv.org/abs/1606.01869

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
