[Supplementary Material]

# Supplementary Material of
# On Robustness of Kernel Clustering

**Bowei Yan**
Department of Statistics and Data Sciences
University of Texas at Austin

**Purnamrita Sarkar**
Department of Statistics and Data Sciences
University of Texas at Austin

## A  Sub-gaussian random vector

In our analysis, we make use of some useful properties of sub-gaussian random variables, which are defined by the following equivalent properties. More discussions on this topic can be found in [1].

**Lemma A.1** ([1]). *The sub-gaussian norm of $X$ is denoted by $\|X\|_{\psi_2}$,*

$$\|X\|_{\psi_2} = \sup_{p \geq 1} p^{-1/2} (\mathbb{E}|X|^p)^{1/p}.$$

*Every sub-gaussian random variable $X$ satisfies:*

*(1) $P(|X| > t) \leq \exp(1 - ct^2/\|X\|_{\psi_2}^2)$ for all $t \geq 0$;*

*(2) $(\mathbb{E}|X|^p)^{1/p} \leq \|X\|_{\psi_2} \sqrt{p}$ for all $p \geq 1$. In particular, $Var(X) \leq 2\|X\|_{\psi_2}^2$.*

*(3) Consider a finite number of independent centered sub-gaussian random variables $X_i$. Then $\sum_i X_i$ is also a centered sub-gaussian random variable. Moreover,*

$$\| \sum_i X_i \|_{\psi_2}^2 \leq C \sum_i \|X_i\|_{\psi_2}^2$$

We say that a random vector $X \in \mathbb{R}^n$ is sub-gaussian if the one-dimensional marginals $\langle X, x \rangle$ are sub-gaussian random variables for all $x \in \mathbb{R}^n$.

We will also see the square of sub-gaussian random variables, the following lemma shows it will be sub-exponential. A random variable is sub-exponential if the following equivalent properties hold with parameters $K_i > 0$ differing from each other by at most an absolute constant factor.

$$P(|X| > t) \leq \exp(1 - t/K_1) \quad \text{for all } t \geq 0; \tag{A.1}$$

$$(\mathbb{E}|X|)^{1/p} \leq K_2 p \quad \text{for all } p \geq 1; \tag{A.2}$$

$$\mathbb{E} \exp(X/K_3) \leq e. \tag{A.3}$$

**Lemma A.2** ([1]). *A random variable $X$ is sub-gaussian if and only if $X^2$ is sub-exponential. Moreover,*

$$\|X\|_{\psi_2}^2 \leq \|X^2\|_{\psi_1} \leq 2\|X\|_{\psi_2}^2$$

We have a Bernstein-type inequality for independent sum of sub-exponential random variables.

**Lemma A.3** ([1]). *Let $X_1, \cdots, X_N$ be independent centered sub-exponential random variable, and $M = \max_i \|X_i\|_{\psi_1}$. Then for every $a = (a_1, \cdots, a_N) \in \mathbb{R}^N$ and every $t \geq 0$, we have*

$$P \left( \left| \sum_{i=1}^N a_i X_i \right| \geq t \right) \leq 2 \exp \left[ -c \min \left( \frac{t^2}{M^2 \|a\|_2^2}, \frac{t}{M\|a\|_\infty} \right) \right]$$

*where $c > 0$ is an absolute constant.*

# B  Proof of Theorem 1

To prove Theorem 1, we work with the elementwise expansion. We use $c$ to represent any constant that does not depend on the parameters, and its value can change from line to line. For $i \in C_k, j \in C_\ell$, recall that $W_i$ is sub-gaussian random vector with mean 0, covariance $\sigma_k^2 I$ and sub-gaussian norm bounded by $b$. We have

$$\|Y_i - Y_j\|_2^2 = \|\mu_k - \mu_\ell\|_2^2 + 2\frac{(W_i - W_j)^T}{\sqrt{p}}(\mu_k - \mu_\ell) + \frac{\|W_i - W_j\|_2^2}{p} \tag{B.1}$$

As $W_i$ and $W_j$ are independent, $W_i - W_j$ has mean 0 and covariance $(\sigma_k^2 + \sigma_\ell^2)I$.

Define

$$\beta_{ij} = \|W_i - W_j\|_2^2/p - (\sigma_k^2 + \sigma_\ell^2),$$
$$\alpha_{ij} = (W_i - W_j)'(\mu_k - \mu_\ell)/\sqrt{p}.$$

Hence $\mathbb{E}\beta_{ij} = 0$. By the Lipschitz continuity of $f$,

$$|K_{ij} - \tilde{K}_{ij}| \le 2C_0|\beta_{ij} + 2\alpha_{ij}| \tag{B.2}$$

By Lemma A.1-(3), $\alpha_{ij}$ is also sub-gaussian, with sub-gaussian norm upper bounded by $2bd_{k\ell}^2 C/p$, for some $C > 0$. Then by Lemma A.1-(1), $\exists C_1 > 0$ s.t.

$$P\left(|\alpha_{ij}| \ge c\sqrt{\frac{\log p}{p}}\right) \le p^{-C_1 c^2} \tag{B.3}$$

$$\beta_{ij} = \sum_{d=1}^p (W_i^{(d)} - W_j^{(d)})^2/p - (\sigma_k^2 + \sigma_\ell^2). \tag{B.4}$$

To bound $\beta_{ij}$, note each summand in Eq. (B.4) is a squared sub-gaussian random variable, thus is a sub-exponential random variable by Lemma A.2. By Lemma A.3 with $t = c\sqrt{\frac{\log p}{p}}$, we see that with $a = (1, \ldots, 1)/p$, $\min\left(c^2\frac{t^2}{M^2\|a\|_2^2}, c\frac{t}{M\|a\|_\infty}\right) = \min\left(\frac{c^2 \log p}{M^2}, \frac{c\sqrt{p \log p}}{M}\right) \ge c'\log p$ for large enough $p$. Thus $\exists C_2 > 0$ such that for large enough $p$,

$$P\left(|\beta_{ij}| \le c\sqrt{\frac{\log p}{p}}\right) \ge 1 - p^{-C_2 c^2} \tag{B.5}$$

By union bound, for some $\rho > 0$, with probability at least $1 - n^2 p^{-\rho c^2}$,

$$\sup_{i,j \in \mathcal{I}} |K_{ij} - \tilde{K}_{ij}| \le c\sqrt{\frac{\log p}{p}}.$$

# C  Proof of Lemma 1

Define a diagonal matrix $D$ where $D_{ii} = f(\sigma_k^2)$, if $i \in C_k$ and 0 if $i \in \mathcal{O}$. Write $\tilde{K}_0 = \tilde{K} - I + D^2$, which is basically replacing the diagonal of $\tilde{K}$ to make it blockwise constant. By the fact $f(d_{k\ell}^2 + \sigma_k^2 + \sigma_\ell^2) = f(d_{k\ell}^2)f(\sigma_k^2)f(\sigma_\ell^2)$, $\tilde{K}_0$ has the decomposition $\tilde{K}_0 = DZBZ^T D$ where $B \in \mathbb{R}^{r \times r}$ and $B_{k\ell} = f(d_{k\ell}^2)$. In fact, $B$ is exactly the Gaussian kernel matrix generated by $\{\mu_i\}_{i=1}^r$ centers, and is strictly positive semi-definite when the scale parameter $\eta \neq 0$ and centers are all different. Hence $\tilde{K}_0$ is rank $r$.

$$\lambda_r(DZBZ^T D) = \lambda_r(B^{1/2}Z^T D^2 ZB^{1/2}) = \lambda_r(BZ^T D^2 Z)$$

The first equality uses the fact that $XX^T$ and $X^TX$ has the same set of eigenvalues. The second step uses the fact that $B$ is full rank, since all clusters have distinct means. Now $B$ and $Z^TD^2Z$ are both $r \times r$ positive definite matrices. So the $r$th eigenvalue is the smallest eigenvalue. Now we use, $\lambda_{\min}(BZ^TD^2Z) \geq \lambda_{\min}(B)\lambda_{\min}(Z^TD^2Z)$ and have

$$\lambda_r(\tilde{K}_0) \geq \lambda_r(Z^TD^2Z)\lambda_r(B) \geq \frac{n}{r}\lambda_{\min}(B) \cdot \min_k \left(f(\sigma_k^2)\right)^2.$$

Then $\lambda_r(\tilde{K}_0) = \Omega(\frac{n}{r})$. On the other hand, $\|I - D^2\|_2 \leq \max_k(1 - f(2\sigma_k^2))$. Let $\lambda_r(\tilde{K}), \lambda_{r+1}(\tilde{K})$ be the $r^{th}$ and $r+1^{th}$ eigenvalue of $\tilde{K}$, by Weyl's inequality,

$$\lambda_r(\tilde{K}) \geq \lambda_r(\tilde{K}_0) - \max_k(1 - f(2\sigma_k^2)) = \Omega(\frac{n}{r}\lambda_{\min}(B))$$

$$\lambda_{r+1}(\tilde{K}) \leq \max_k(1 - f(2\sigma_k^2)) = O(1) \tag{C.1}$$

Putting pieces together,

$$\lambda_r(\tilde{K}) - \lambda_{r+1}(\tilde{K}) \geq \frac{n}{r}\lambda_{\min}(B) \cdot \min_k \left(f(\sigma_k^2)\right)^2 - 2\max_k(1 - f(2\sigma_k^2)) = \Omega\left(\frac{n}{r}\lambda_{\min}(B)\right).$$

# D  Proof of Lemma 2

*Proof.* First note that $\hat{X}$ is the optimal solution of (SDP-1), so $\langle K, \hat{X} \rangle \geq \langle K, X_0 \rangle$. Hence $\langle K - \tilde{K}, \hat{X} - X_0 \rangle \geq \langle \tilde{K}, X_0 - \hat{X} \rangle$.

Let $a := \min_k f(2\sigma_k^2)$, $b := \max_{k \neq \ell} f(d_{k\ell}^2 + \sigma_k^2 + \sigma_\ell^2)$ and $\gamma_{\min} := a - b$, we have

$$\begin{aligned}
\langle \tilde{K}, X_0 - \hat{X} \rangle &= \sum_k \sum_{i \in \tilde{C}_k} \left( \sum_{j \in \tilde{C}_k} f(2\sigma_k^2)(1 - \hat{X}_{ij}) - \sum_{\ell \neq k} \sum_{j \in \tilde{C}_\ell} f(d_{k\ell}^2 + \sigma_k^2 + \sigma_\ell^2)\hat{X}_{ij} \right) \\
&\geq \sum_k \sum_{i \in \tilde{C}_k} \left( a \sum_{j \in \tilde{C}_k} (1 - \hat{X}_{ij}) - b \sum_{\ell \neq k} \sum_{j \in \tilde{C}_\ell} \hat{X}_{ij} \right) \\
&\geq \sum_k \sum_{i \in \tilde{C}_k} \left( a \sum_{j \in \tilde{C}_k} (1 - \hat{X}_{ij}) - b \left( \frac{n}{r} - \sum_{j \in \tilde{C}_k} \hat{X}_{ij} \right) \right) \\
&\geq \gamma_{\min} \sum_k \sum_{i \in \tilde{C}_k} \sum_{j \in \tilde{C}_k} (1 - \hat{X}_{ij})
\end{aligned} \tag{D.1}$$

On the other hand, by the fact that $\hat{X}_{ij} \geq 0$ and row sum is $n/r$,

$$\begin{aligned}
\|X_0 - \hat{X}\|_1 &= \sum_k \sum_{i \in \tilde{C}_k} \left( \sum_{j \in \tilde{C}_k} (1 - \hat{X}_{ij}) + \sum_{\ell \neq k} \sum_{j \in \tilde{C}_\ell} \hat{X}_{ij} \right) \\
&= \sum_k \sum_{i \in \tilde{C}_k} \left( \sum_{j \in \tilde{C}_k} (1 - \hat{X}_{ij}) + \left( n/r - \sum_{j \in \tilde{C}_k} \hat{X}_{ij} \right) \right) \\
&\leq 2 \sum_k \sum_{i \in \tilde{C}_k} \sum_{j \in \tilde{C}_k} (1 - \hat{X}_{ij})
\end{aligned} \tag{D.2}$$

Equations (D.1) and (D.2) gives us:

$$\|X_0 - \hat{X}\|_1 \leq \frac{2}{\gamma_{\min}} \langle \tilde{K}, X_0 - \hat{X} \rangle \leq \frac{2\langle K - \tilde{K}, \hat{X} - X_0 \rangle}{\gamma_{\min}}$$

$\square$

# E    Proof of Theorem 2

By Lemma 2,

$$\|X_0 - \hat{X}\|_1 \leq \frac{2\langle \tilde{K}, X_0 - \hat{X}\rangle}{\gamma_{\min}} \leq \frac{2\langle K - \tilde{K}, \hat{X} - X_0\rangle}{\gamma_{\min}}$$

Divide the inner product into inlier part and outlier part, and note that $0 < |K_{ij} - \tilde{K}_{ij}| < 1, \forall i, j$. By Theorem 1, w.p. at least $1 - n^2 p^{-\rho c^2}$, we have

$$
\begin{aligned}
&\langle K - \tilde{K}, \hat{X} - X_0\rangle \\
&= \sum_{(i,j)\in\mathcal{I}\times\mathcal{I}} (K_{ij} - \tilde{K}_{ij})(\hat{X}_{ij} - (X_0)_{ij}) + \sum_{(i,j)\in\mathcal{R}} (K_{ij} - \tilde{K}_{ij})(\hat{X}_{ij} - (X_0)_{ij}) \\
&\leq \|\hat{X} - X_0\|_1 \cdot \|K^{\mathcal{I}\times\mathcal{I}} - \tilde{K}^{\mathcal{I}\times\mathcal{I}}\|_\infty + \sum_{(i,j)\in\mathcal{R}} (\hat{X}_{ij} - (X_0)_{ij})(K_{ij} - \tilde{K}_{ij}) \\
&\leq \|\hat{X} - X_0\|_1 \cdot \|K^{\mathcal{I}\times\mathcal{I}} - \tilde{K}^{\mathcal{I}\times\mathcal{I}}\|_\infty + \sum_{(i,j)\in\mathcal{R}} \hat{X}_{ij}(K_{ij} - \tilde{K}_{ij}) - \sum_{(i,j)\in\mathcal{R}} (X_0)_{ij}(K_{ij} - \tilde{K}_{ij}) \\
&\leq \|\hat{X} - X_0\|_1 \cdot \|K^{\mathcal{I}\times\mathcal{I}} - \tilde{K}^{\mathcal{I}\times\mathcal{I}}\|_\infty + \sum_{(i,j)\in\mathcal{R}} \hat{X}_{ij} + \sum_{(i,j)\in\mathcal{R}} (X_0)_{ij} \\
&\leq C\sqrt{\frac{\log p}{p}}\|X_0 - \hat{X}\|_1 + \frac{4mn}{r}
\end{aligned}
$$

Thus,

$$\left(\gamma_{\min} - 2C\sqrt{\frac{\log p}{p}}\right)\|\hat{X} - X_0\|_1 \leq \frac{4mn}{r}$$

When $\sqrt{\frac{\log p}{p}} = o(\gamma_{\min})$, rearranging terms gives

$$\|X_0 - \hat{X}\|_1 \leq \frac{\frac{4mn}{r}}{\gamma_{\min} - C\sqrt{\frac{\log p}{p}}} \leq \frac{4mn}{r\gamma_{\min}}\left(1 + \frac{C'}{\gamma_{\min}}\sqrt{\frac{\log p}{p}}\right) = O\left(\frac{mn}{r\gamma_{\min}}\right)$$

# F    Davis-Kahan Theorem

**Theorem F.1** ([2]). *Let $\Sigma, \hat{\Sigma} \in \mathbb{R}^{p\times p}$ be symmetric, with eigenvalues $\lambda_1 \geq \cdots \geq \lambda_p$ and $\hat{\lambda}_1 \geq \cdots \geq \hat{\lambda}_p$ respectively. Fix $1 \leq r \leq s \leq p$ and assume that $\min(\lambda_{r-1} - \lambda_r, \lambda_{s-1} - \lambda_s) > 0$, where $\lambda_0 := \infty$ and $\lambda_{p+1} := -\infty$. Let $d := s - r + 1$, and let $V = (v_r, v_{r+1}, \cdots, v_s) \in \mathbb{R}^{p\times d}$ and $\hat{V} = (\hat{v}_r, \hat{v}_{r+1}, \cdots, \hat{v}_s) \in \mathbb{R}^{p\times d}$ have orthonormal columns satisfying $\Sigma v_j = \lambda_j v_j$ and $\hat{\Sigma}\hat{v}_j = \hat{\lambda}_j\hat{v}_j$, for $j = r, r+1, \cdots, s$. Then there exists an orthogonal matrix $\hat{O} \in \mathbb{R}^{d\times d}$ such that*

$$\|\hat{V}\hat{O} - V\|_F \leq \frac{2^{3/2}\|\hat{\Sigma} - \Sigma\|_F}{\min(\lambda_{r-1} - \lambda_r, \lambda_{s-1} - \lambda_s)}.$$

# G    Proof of Lemma 3

We prove the result for k-means on $\hat{X}$. Let $\hat{U}$ be the top $r$ eigenvectors of $\hat{X}$, $U \in \mathbb{R}^{n\times r}$ be the top $r$ eigenvector of $X_0$, then by construction, it can be written as $U = \begin{bmatrix} U^{\mathcal{I}} \\ U^{\mathcal{O}} \end{bmatrix}$. Let $\nu \in \mathbb{R}^{r\times r}$ be the population value of the eigenvector corresponding to each cluster, $U = Z\nu$. $U$ is a unit basis so we know $I = U^T U = \nu^T Z^T Z\nu = \frac{n}{r}\nu^T\nu$. So $\nu^T\nu = \frac{r}{n}I_r$.

Define $\mathcal{C} = \{M \in \mathbb{R}^{n \times r} : M \text{ has no more than } r \text{ unique rows}\}$. Then minimizing the k-means objective for $\hat{U}$ is equivalent to

$$\min_{\{m_1, \cdots, m_r\} \subset \mathbb{R}^r} \sum_i \min_g \|\hat{u}_i - m_g\|_2^2 = \min_{M \in \mathcal{C}} \|\hat{U} - M\|_F^2$$

So $C = [c_1, \cdots, c_n] = \arg\min_{M \in \mathcal{C}} \|\hat{U} - M\|_F^2$ and $\|C - \hat{U}\| \leq \|Z\nu O - \hat{U}\|$. $c_i$ is the center assigned to point $i$ by running k-means on $\hat{U}$.

When $i, j \in \mathcal{I}, Z_i \neq Z_j$,

$$\|Z_i \nu - Z_j \nu\| = \|(Z_i - Z_j)\nu\| \geq \sqrt{2} \min_{x:\|x\|^2=1} \sqrt{x^T \nu^T \nu x} = \sqrt{\frac{2r}{n}}$$

So

$$\|c_i - Z_j \nu O\|_2 \geq \|Z_i \nu - Z_j \nu\| - \|c_i - Z_i \nu O\| \geq \sqrt{\frac{2r}{n}} - \sqrt{\frac{r}{2n}} = \sqrt{\frac{r}{2n}} \tag{G.1}$$

Therefore when $i, j \in \mathcal{I}$ and $Z_i \neq Z_j$, $\|c_i - Z_i \nu O\| < \sqrt{\frac{r}{2n}} \Rightarrow \|c_i - Z_i \nu O\|_2 < \|c_i - Z_j \nu O\|_2$, which means node $i$ is correctly clustered.

Now we bound the cardinality of $\mathcal{M}$.

$$|\mathcal{M}| \leq \frac{2n}{r} \sum_{i \in \mathcal{I}} \|c_i - Z_i \nu O\|_F^2 = \frac{2n}{r} \|C^{\mathcal{I}} - U^{\mathcal{I}} O\|_F^2$$

$$\leq \frac{2n}{r} (\|C^{\mathcal{I}} - \hat{U}^{\mathcal{I}}\|_F + \|\hat{U}^{\mathcal{I}} - U^{\mathcal{I}} O\|_F)^2$$

$$\|C^{\mathcal{I}} - \hat{U}^{\mathcal{I}}\|_F^2 = \|\hat{U} - C\|_F^2 - \|C^{\mathcal{O}} - \hat{U}^{\mathcal{O}}\|_F^2$$

$$\leq \|\hat{U} - C\|_F^2 \leq \|\hat{U} - UO\|_F^2$$

Therefore,

$$|\mathcal{M}| \leq \frac{2n}{r} (\|\hat{U} - UO\|_F + \|\hat{U}^{\mathcal{I}} - U^{\mathcal{I}} O\|_F)^2 \leq \frac{8n}{r} \|\hat{U} - UO\|_F^2$$

For k-means procedure on $K$, note that $\tilde{K}$ is blockwise constant except for the diagonals. It can be shown that the top $r$ eigenvectors of $\tilde{K}$ are also piecewise constant. The rest of the analysis is similar to that of $\hat{X}$.

## H   Proof of Corollary 1

*Proof.* Denote by $d_0$ the distance between clusters, $\alpha = f(2\sigma^2), \beta = f(d_0^2 + 2\sigma^2)$, hence $\gamma_{\min} = \alpha - \beta$. Then $\tilde{K}$ has the form $(\alpha - \beta)X_0 + \beta E + (1 - \alpha)I$, and $\lambda_r(\tilde{K}) \geq \gamma_{\min} n/r$, since $\beta E + (1 - \alpha)I$ is positive semidefinite.

On the other hand, from Lemma 1 and Eq. (C.1), $\lambda_{r+1}(\tilde{K}) \leq 1 - f(2\sigma^2) \leq 1$. Hence $\lambda_r - \lambda_{r+1} \geq \frac{n}{r} \gamma_{\min} - 1$. By Lemma 3 the misclassification rate of K-SVD becomes:

$$|\mathcal{M}_{ksvd}| \leq C\frac{n}{r} \left( \frac{2^{3/2} \|\tilde{K} - K\|_F}{\lambda_r(\tilde{K}) - \lambda_{r+1}(\tilde{K})} \right)^2$$

$$\leq C\frac{n}{r} \left( \frac{\max\left\{ n\sqrt{\frac{\log p}{p}}, \sqrt{mn} \right\}}{\frac{n}{r} \gamma_{\min}} \right)^2$$

$$\leq \max \left( O_P \left( \frac{mr}{\gamma_{\min}^2} \right), O_P \left( \frac{nr \log p/p}{\gamma_{\min}^2} \right) \right)$$

$\square$