[Reviews · NeurIPS 2016]

Reviewer 1

Summary

This paper shows that in the context of kernel clustering, an SDP relaxation provides strong consistency and better performance with respect to outliers when compared to an SVD relaxation.

Qualitative Assessment

Although the main findings and analysis look promising and it is interesting for the community, I'm concerned about the following: Assumptions: * It is very unrealistic to assume that all clusters have the same size. What happens if this assumption is lifted ? * The number of clusters k is known. This is also unrealistic since in most real-life problems, k is not known. * The centroids are well-separated. Again this casts doubts on whether the analysis is valid in real-life problems. Experiments: * All experiments are performed using toy problems and in some cases, the parameters seem to be set arbitrarily. * Same \sigma, same number of clusters k. These parameters are critical in kernel clustering and the current state of the experiments fails to convince the reader.

Confidence in this Review

2-Confident (read it all; understood it all reasonably well)


Reviewer 2

Summary

This paper introduces a semidefinite programming relaxation for the kernel clustering problem, then proves that under a suitable model specification, both the Kernel-SVD and Kernel-SDP approaches are consistent in the limit, albeit SDP is strongly consistent, i.e. achieves exact recovery, whereas K-SVD is weakly consistent. Experimental results on toy datasets are presented.

Qualitative Assessment

Here are few comments. 1. The analysis assumes that all inlier clusters are of equal size and uses this fact for analysis. However, this seems to be a very restrictive assumption. In real life, it is very unlikely that all clusters will be of equal size. It will be nice to have a discussion describing how much additional work is necessary to remove this unrealistic assumption. 2. Using the current notation both the algorithms takes only inliers as input (Y-1,... Y_n are inliers). It will be nice to change the notation so that input to both algorithms can be both inliers as well as outliers. 3. Spectral clustering is based on normalized/unnormalized "Laplacian matrix" and not kernel matrix. It will be nice to have a discussion how they are related. Note that in case of spectral clustering using Laplacian matrix, eigenvectors corresponding to the lowest eigenvalues of Graph laplacian is used for embedding and k-means clustering, where as in case of kernel-PCA eigenvectors corresponding to the largest eigenvalues of the Gram matrix are used for k-means clustering. 4. Theorem 1 is based on Gaussian kernel (line 135) so is Lemma 1. Therefore, notation wise it will be easier to use K, \tilde{K} and exponential in place of function f etc. 5. The theoretical results of this paper are " high probability" results based on theorem 1. Theorem 1 uses an union bound and the failure probability is n^2p^{-c^2\rho}, where c and \rho are constants. Note that c is controllable depending on how much deviation one wants but \rho is just an existential constant. Therefore, for fixed data dimension p, as n tends to infinity the failure probability tends to one. For any fixed p, this limits the maximum size of inliers that the algorithm can handle. In particular if we want a failure probability of \delta, that implies n can be at most \sqrt{\delta p^{c^2\rho}}$. 6. Statement of Lemma 1 is a little unclear. In particular, the inequality in the proof is not clear. Note that only a lower bound of the eigengap is proved. No upper bound Big-Oh is proved and therefore using \Theta notation may not be correct. 7. In the supplementary material (proof of Lemma 1), no justification is not provided for the inequality between line 67 and 68. It is not clear how this inequality is obtained. 8. In proof of theorem 3, first only inliers are considered when no outliers are present and Davis-Kahan theorem is applied. However, in this case the corresponding expression, equation 4, involves $m$ which is the number of ouliers. This sounds strange because this analysis does not consider any outlier and since presence of ouliers are handled right after equation 4 and is given in equation 5. 9. Proof of theorem 3 utilizes results of [22] which requires that centroids are well separated. It guaranteed in proof of theorem 3 that the centroids are well separated so that results of [22] can be applied. 10. The accuracy of the clustering process is measured based on the criteria that inliers are assigned to the correct respective clusters. However, in presence of outliers, if an outlier is assigned to a cluster (consists of inliers) that is also an error. How is this error taken into consideration in the analysis? 11. All experimenal results are on toy datasets involving same \sigma for all clusters. Experimental results can be improved by considering, (a) different sigma values for different clusters in case of toy (synthetic) datasets, (b) varying dimensionality p for synthetic datasets (choce of specific p=160 seems arbitrary), (c) applying the proposed method on realworld datasets.

Confidence in this Review

2-Confident (read it all; understood it all reasonably well)


Reviewer 3

Summary

In the paper the robustness of kernel K-means is analyzed and performance guarantees are given for sub-Gaussian distributions for sparse cluster means and in presence of outliers. The results are supported by proofs.

Qualitative Assessment

some details are not made explicit in the paper such as the mutual convergence of p/n. If p/n->0 as is often assumed in classical statistics, [11] does not apply here since it requires p/n --> c != 0 which is another regime not fulfilling some basic convergence theorems of classical statistics. The authors declare in which asymptotic regime they are operating to make clear for which kinds of parameters their theorems hold. At several points the authors should be more explicit about the variables and the terminology used. I had to look up the meaning of some variables in the additional material. Instead of writing: T(n) is O(n) iff... it should be something like: A function T(n) is O(n) iff... I was confused and started looking for the definition of T(n) first :) I don't see why the affinity matrix in Assumption 1 is also a Gram matrix. A Gram matrix is defined as G := X^TX. Hence, this would apply to the inner product kernel only. The notation is introduced at two places in 2 (at the very beginning and the very end). You should make it appear at only one place so that a reader nows where to look it up. The discussion of the asymptotic regime should definitely be fixed before publication. Apart from that it seems to be a very favourable and mathematically rigorous work with comprehensible proofs. The language is very clear and free of mistakes.

Confidence in this Review

2-Confident (read it all; understood it all reasonably well)


Reviewer 4

Summary

As claimed, semidefinite programming relaxation is, at the first time, used to do kernel clustering. It is shown that proposed algorithm achieves better consistency and robustness in some sense than kernel SVD algorithms.

Qualitative Assessment

Overall looks good. The authors did shed lights on the robustness of kernel clustering algorithms. The following are a few questions I have: Line 138: wouldn't it make more sense to define $\tilde{K}_f as f(\sigma_k^2+\sigma_l^2) if i=j$? Line 156: more justification for the positiveness of $r_{min}$ Line 161: the concentration of $K$ from Theorem 1 is for a submatrix of $K$ (corresponding to the index set $I$), how to extend the results for $K$? Figure 1: I saw some potential contradictions between (b) and (c). If we focus on the performace of SDP, in (b) it shows the accuracy is roughly 0.9 when m=60 and d^2=0.08, while in (c) it shows the accuracy is roughly 0.45 with the same parameters. So what changes here? Plus, there should be more explanations about the parameters. For example, what is $d^2$ (I assume it is min distance between centers)? Moreover, wouldn't it be more persuasive if more experiments are conducted on some real data sets?

Confidence in this Review

1-Less confident (might not have understood significant parts)


Reviewer 5

Summary

The robustness of kernel clustering has not been sufficiently explored yet. The paper proposed two methods for kernel clustering, i.e. the semidefinite programming relaxation for the kernel clustering problem (SDP) and kernel singular value decomposition (K-SVD), and investigated the consistency and robustness of two kernel-based clustering algorithms.

Qualitative Assessment

1. K-PCA is a technique for dimensionality reduction, it is ambiguous that the paper considers it as a kernel clustering algorithm without any extra stating (line 101). 2. It seems that the paper applies a new SDP relaxation to the kernel clustering for the first time according to the author(s), but just reminds there exist the same relaxation in previous work (line 107), lacking of explanation for why it is reasonable and effective to do so. 3. The paper does not give analysis on robustness of other kernel clustering methods based on semidefinite relaxation (if there exist such method), thus lacking of comparision to show the essentialness for proposing the SDP relaxation for kernel clustering. 4. The experimental results on synthetic data set seem not to be convincing, tests on real data sets are needed. 5. Some terms and notations may be ambiguous without special explanation, e.g. "population matrix" (line 136), "[n+m]" (line 63).

Confidence in this Review

2-Confident (read it all; understood it all reasonably well)


Reviewer 6

Summary

The authors provide a robustness analysis of two popular forms of kernel k-means clustering: a semi-definite programming relaxation and a kernel singular value decomposition. The goal is to understand behavior in the presence of (a possibly growing number of) outliers, in particular, consistency. They adopt a problem set-up that is well-matched to k-means (e.g. equal-sized clusters) and partition the data into groups of inliers and outliers. The framework is also well suited for scenarios where the data lie on a lower-dimensional manifold that may be obscured. Their theoretical analysis and results are accompanied by a very nice summary, one that will serve audiences not as familiar with the technical details of the proofs. In short, SDP is the winner, showing strong consistency when there are no outliers and also better performance when the number of outliers grows slowly as compared to the number of observations. The numeric analyses show general agreement with the theoretical results.

Qualitative Assessment

In general, I thought the paper was well-written and made some nice theoretical contributions to this area. What I valued most was the authors' ability to explain the theoretical results conceptually along side the technical details. This is hard to do and not that common. The paper's audience and influence will be larger for it. I would have liked to see some more discussion about the assumption that the noise is sub-gaussian and isotropic. Something along the lines of how limiting those assumptions might be in practice and if some of their results might still apply (via some sort of approximation). In the experiments, the authors use an extra cluster for outliers, but it seems odd to assume that the outliers would have a similar distribution or structure as the actual clusters. In mixture model clustering, it is common to assume something like a uniformly distributed cluster to pick up the noise/outliers or to use a contaminated distribution to pick up outliers. Treating all the outliers like they're a group of the same size and distribution as the rest of the inliers sounds unrealistic. p.2, line 76, should be "where the data lie on a low-dimensional manifold" p.3, line 111, should be "One first does SVD on …, then does k-means on the first k eigenvectors" p.5, lines 182/183 - inconsistency in algorithm notation p.5, line 185 - K-means should be capitalized (it is on p.7) p.8, line 275, should be "K-SVD is doing almost as well" p.8, lines 285-286, should be "as the number of observations grows and the outliers are a small fraction of the inliers"

Confidence in this Review

2-Confident (read it all; understood it all reasonably well)